# Optimization of Manganese Recovery from a Solution Based on Lithium-Ion Batteries by Solvent Extraction with D2EHPA

**Nathália Vieceli [1,*], Niclas Reinhardt [2], Christian Ekberg [1] and Martina Petranikova [1]**

[1] Department of Chemistry and Chemical Engineering, Industrial Materials Recycling and Nuclear Chemistry, Chalmers University of Technology, SE-41296 Gothenburg, Sweden; che@chalmers.se (C.E.); martina.petranikova@chalmers.se (M.P.)

[2] MEAB Metallextraktion AB, Datavägen 51, S-43632 Askim, Sweden; niclas@meab-mx.se

* Correspondence: nathalia.vieceli@chalmers.se

**Abstract:** Manganese is a critical metal for the steelmaking industry, and it is expected that its world demand will be increasingly affected by the growing market of lithium-ion batteries. In addition to the increasing importance of manganese, its recycling is mainly determined by trends in the recycling of iron and steel. The recovery of manganese by solvent extraction has been widely investigated; however, the interaction of different variables affecting the process is generally not assessed. In this study, the solvent extraction of manganese from a solution based on lithium-ion batteries was modeled and optimized using factorial designs of experiments and the response surface methodology. Under optimized conditions (O:A of 1.25:1, pH 3.25, and 0.5 M bis(2-ethylhexyl) phosphoric acid (D2EHPA)), extractions above 70% Mn were reached in a single extraction stage with a coextraction of less than 5% Co, which was mostly removed in two scrubbing stages. A stripping product containing around 23 g/L Mn and around 0.3 g/L Co can be obtained under optimized conditions (O:A of 8:1, 1 M $H_2SO_4$ and around 13 min of contact time) in one stripping stage.

**Keywords:** lithium-ion battery; battery recycling; manganese recovery; solvent extraction; D2EHPA; factorial design of experiments



## 1. Introduction

Manganese is one of the most abundant metals in the Earth's crust; however, manganese is highly dispersed (low-grade), and minerals are widely distributed. The identified manganese resources are concentrated in a few countries—the main manganese mining areas are in China, South Africa, Australia, and Gabon [1–3].

The main end use of manganese is in the steel industry, which accounts for 90% of the world´s manganese demand. Manganese is also widely used in ironmaking and alloys with aluminum, magnesium, and copper [3–6]. Non-metallurgical applications account for only 5–10% of the manganese consumption, which is used in electrical systems, in the chemical industry, in the ceramic and glass production, and in the agricultural sector [7]. In electrical systems, manganese dioxide is used for cathodic depolarizer in dry cells, alkaline batteries, and lithium-ion batteries (LIBs) [4].

Natural manganese dioxide is used in dry cells, while high-grade synthetic manganese dioxide is produced chemically or by electrolysis to be used in alkaline batteries and LIBs [4]. Lithium manganese spinels (such as $LiMn_2O_4$) and layered lithium–nickel–manganese–cobalt (NMC) oxide systems have an important role in the development of advanced rechargeable lithium-ion batteries, with cost and environmental advantages [8]. Thus, nowadays, most automakers and some electronics makers use some version of NMC system in their LIBs [9].

In this context, the United States of America Department of Defense has recently classified manganese as one of the most critical mineral commodities for the United States because it is essential for important industrial sectors, has no substitutes, and has a

potential for supply disruptions, since the country is strongly dependent on imports [10]. Additionally, the United States included electrolytic manganese metal in the National Defense Stockpile in 2019 as a critical material for defense purposes [2].

Although it is expected that steel will continue leading the manganese demand, the consumption of manganese in batteries applications is projected to grow fast in the next decade, boosted by the rapid growth in the lithium-ion battery market, which is expected to increase from $35 billion USD in 2020 and reach $71 billion USD in 2025 [11,12]. Thus, electrolytic manganese dioxide (EMD) for the battery industry is expected to be the fastest-growing segment of the manganese market [13], increasing the manganese production along with the global demand for batteries [14].

EMD is generally produced from high-grade manganese ores [15], and in general, converting manganese ores to EMD involves a high-temperature pyrometallurgical process, which has some drawbacks such as environmental impacts, high-energy consumption, and high costs. Furthermore, because the roasting process decreases the oxide content in the ore, EMD producers face competition from chemical and steel industry buyers of high-grade manganese ores [16]. In this context, the recovery of manganese from spent LIBs can help decrease supply risks and impacts linked to the primary production of manganese. However, although there is an increasing importance of manganese, its recycling is mainly determined by trends in the recycling of iron and steel, and in general, materials are not recycled specifically due to their manganese content [2,17,18]. Moreover, when it comes to LIBs recycling, the presence of manganese in the leaching solutions has been linked to a decrease in the selective separation of cobalt and nickel, and for this reason, manganese should be previously recovered [19,20].

The recovery of manganese from primary and secondary resources by solvent extraction has been investigated by several authors [14,20–26]. Table 1 (on the next page) summarizes the optimal extraction conditions described in some studies focused on the extraction of manganese from different feed solutions, including from leach solutions from spent LIBs. It is possible to highlight that bis(2-ethylhexyl) phosphoric acid (D2HEPA) is the most widely used extractant to recover Mn from liquors from LIBs as well as from other solutions.

Although several studies on the recovery of manganese by solvent extraction have been published, the effect of different variables affecting the process is generally approached using one-factor-at-a-time, which does not allow identifying interaction effects among them. In this context, the main goal of this study was to optimize the solvent extraction of manganese using the factorial design of experiments and response surface methodologies to assess and model the effects of the variables affecting the process. The optimization of the recovery of manganese was studied using a synthetic solution based on an acid leach from spent LIBs. The results can support further investigations focused on the recovery of manganese from spent LIBs, which can be considered an important secondary resource of a critical material for many important industrial sectors.

**Table 1.** Summary of conditions for the manganese solvent extraction from published studies (corresponding to the best conditions reported).

| Extractant | Saponification | Modifier | O:A | Optimum pH | Temperature (°C) | Contact Time (min) | Feed | Initial Composition (g/L) | | | | %E (Mn) | Reference |
|---|---|---|---|---|---|---|---|---|---|---|---|---|---|
| | | | | | | | | Mn | Co | Ni | Li | | |
| 0.4 M D2EHPA | - | - | 2:1 | 3.2 | 25 | 15 | Leach solution produced from spent LIBs (acid leaching with H$_2$SO$_4$ and H$_2$O$_2$) | 3.66 | 19.33 | 5.19 | 3.58 | - | [20] |
| 15% D2EHPA | 60% (with 0.5 M ammonia) | 5% TBP | 1:1 | 2.25 | 25 | 5 | Leaching liquor of spent LIBs | 5.91 | 24.79 | 6.24 | 6.68 | 99.9 | [26] |
| 4 M (D2EHPA/Mn molar) | 65% (with NaOH 5 M) | 10% TBP | 1:1 | 3.8 | room | 10 | Electrodic LIB powder pre-leached with H$_2$SO$_4$ | 4.6 | 21.8 | 2.7 | 3.2 | ~90 | [25] |
| 0.05 M NaD2EHPA (best results) | - | 5% TBP | 9:8 | 2.7 | 30 | 5 | Stock solution with Mn and Co (0.01 M) | 0.01 M | 0.01 M | - | - | 99.94% | [27] |
| 15% Cobalt loaded D2EHPA | 70–75% (with NaOH 10 M) | 5% TBP | 1:1 | 3.2 | 25 | 5 | Sulfuric acid leaching liquor of mixed types of cathode materials (real sample) waste cathode materials | 6.31 | 6.45 | 6.89 | 1.6 | 99% | [24] |
| 20% PC88A/25% Versatic 10 | - | - | 1:1 | 4.5 | room | 5 | Leaching solution from spent LIBs | 11.7 | 11.4 | 12.2 | 5.3 | 99.5% | [23] |
| 25% Cobalt loaded D2EHPA | - | 1-decanol | 1:1 | 3.5 | 25 | 5 | Cobalt electrolyte solution | 0.8 | 55.7 | - | - | 100% (70% in one stage) | [28] |
| 10% D2EHPA | - | 5%TBP | 1:1 | 3.5 | 40 | 10 | Synthetic laterite solution containing Ni, Co, Mn, Mg, Zn, and Cu | 2 | 0.3 | 3 | - | 99% | [29] |

**Table 1.** *Cont.*

| Extractant | Saponification | Modifier | O:A | Optimum pH | Temperature (°C) | Contact Time (min) | Feed | Initial Composition (g/L) | | | | %E (Mn) | Reference |
|---|---|---|---|---|---|---|---|---|---|---|---|---|---|
| | | | | | | | | Mn | Co | Ni | Li | | |
| 30% D2EHPA | 20% (with NaOH 10 M) | 5% TBP | 1:1 | 2.6–2.7 | room | 15 | Leaching solution from spent LIBs, treated with $H_2SO_4$ and $H_2O_2$ | 2 | 0.3 | 3 | - | Removal of Mn and Cu | [22] |
| 40% D2EHPA | - | - | 1:1 | 3.5 | room | 10 | Leaching acid solution from cathode material | 9.18 | 11.32 | 11.51 | 1.76 | ~100 | [21] |
| 0.4 M D2EHPA | - | - | 2:1 | 3.2 | 25 | 15 | Leach solution produced from spent LIBs (acid leaching with $H_2SO_4$ and $H_2O_2$) | 3.66 | 19.33 | 5.19 | 3.58 | - | [20] |
| 20% D2EHPA (0.6 M) | 70–75% (with NaOH 10 M) | - | 1:2 | 4–5 | 25 | 5 | Co, Ni, and Li were removed by precipitation | 5.27 | 5.84 | 4.93 | 1.25 | 97% | [30] |
| D2EHPA | - | - | - | 2.5–3.5 | n.i. | n.i. | Leach liquor from LIBs | 5–30 | 5 to 45 | 5 to 30 | 1 to 10 | 100% | [31] |
| 25% D2EHPA (Cyanex 272 was also tested) | - | - | 1:1.5 | 2.7 | 5 and 25 | n.i. | Synthetic sulfuric acid solutions (Ca, Mn, Na, and Mg) | 0.58–5.3 | - | - | - | 65% | [32] |
| D2EHPA | - | - | 1:1–1:5 | 2.2–2.3 | 40 | continuous | Kakanda tailings (Cu and Co recovery in RDC) | 1.3 | 3 | - | - | 70–90% | [33] |
| 20% D2EHPA | - | - | 1:1 | 2.2–2.3 | n.i. | continuous | Cobalt bearing feed from a cobalt refinery in South Africa. Fe and Cu were first precipitated | 0.1 | 5.5 | - | - | 100% | [34] |

n.i.: not informed.

## 2. Materials and Methods

Bis(2-ethylhexyl) phosphoric acid (D2EHPA, 97%, Sigma Aldrich, Germany) was used as solvent extraction reagent as it was supplied, without any additional purification. Isopar L (Exxon Mobil, USA) was used as diluent. A synthetic solution was prepared based on the chemical composition of an original solution obtained through the acid leaching of spent lithium-ion batteries with sulfuric acid, which was investigated in detail in previous work (unpublished results). The synthetic solution was prepared using sulfates ($NiSO_4.6H_2O$, $CoSO_4.7H_2O$, $MnSO_4.H_2O$, $Li_2SO_4$, Sigma Aldrich, Germany) and Milli-Q water. Impurities typically present in acid leach solutions from LIBs such as Cu and Al were not included into the synthetic solution because they are generally removed using conventional purification processes, for example, cementation and purification, before the solvent extraction.

Preliminary extraction tests, scrubbing, and stripping tests were performed in glass vials (3.5 mL) using a shaking machine (IKA-Vibrax, Germany) operating with 1000 vibrations per min to promote the contact between phases. The experiments were performed at room temperature. Specific conditions used in the preliminary tests are reported in the Results section. The extraction and stripping of manganese and cobalt were optimized using factorial designs of experiments and response surfaces. These methodologies are explained in detail by Montgomery [35]. For the factorial design of experiments of the extraction phase, tests were carried out using plastic containers (50 mL), in which the stirrer from a mixer-settler device was coupled. The stirring speed was set at 1000 rpm, and the tests were also performed at room temperature.

The pH of the aqueous phase was measured using a pH meter (Metrohm 827 pH lab, Switzerland), and the electrode was regularly calibrated before and during the experimental procedures. The pH was adjusted whenever it was needed with 5 M or 10 M NaOH to minimize the dilution effect of the feed solution. Samples from the aqueous phase were taken 10 min after finishing the contact time at the established pH to obtain a complete separation of phases. Chemical analysis was performed by Inductively Coupled Plasma—Optical Emission Spectroscopy (ICP-OES, iCAP™ 6000 Series, USA) using samples from the aqueous phase, which were diluted in 0.5 M nitric acid. The extraction efficiency of metals was determined by Equation (1):

$$\%E = 100 * \frac{D_X}{D_X + (V_{aq}/V_{org})} \tag{1}$$

where $V_{aq}$ and $V_{org}$ represent the volume of the aqueous phase and the volume of the organic phase, respectively, and $D_X$ is the distribution ratio, which describes the ratio between the concentration of a certain metal ($X$) in the aqueous phase and in the organic phase and it can be determined by Equation (2). In some cases, the log $D$ is used to assist the interpretation of results.

$$D_x = C_{X\ organic}/C_{X\ aqueous} \tag{2}$$

The separation factors ($\beta$) between two elements ($X$ and $Y$) can be calculated using Equation (3), and it is determined by the division of the distribution ratio of each element, being normally greater than one. This equation was used to determine the separation factor of manganese in preference to other metals.

$$\beta = D_X/D_Y \tag{3}$$

*Experimental Design*

A full $2^k$ factorial design of experiments was used to fit a second-order linear regression model to the experimental results. To estimate the experimental uncertainty, four additional experiments were performed under the same conditions at the central level of the factors ($n_C$, central point). The effects of three factors (k = 3), each one with two levels ($2^3$ factorial

design), on the process response (*y*, manganese extraction or cobalt extraction) were studied. The factors and levels were selected based on results from preliminary tests and on the literature review.

Experimental design of the extraction stage: The factors investigated in the design of experiments to model the extraction stage were equilibrium pH ($x_1$), organic to aqueous ratio, O:A ($x_2$) and molar concentration of D2EHPA ($x_3$). Each factor was varied in two levels.

Experimental design of the stripping stage: To model the stripping stage, the effect of the following three factors was evaluated: molar concentration of sulfuric acid ($x_1$), organic to aqueous ratio, O:A ($x_2$) and stripping time ($x_3$). Each factor was varied in two levels.

Axial points were included (*2k* axial points) in both designs to estimate the quadratic terms of the models, setting up a central composite design. Tests were performed in random order. The distance of the axial points from the central point was $\alpha = 1$ (face-centered central composite design). The standard, high, and low levels of the factors are presented in Table 2.

**Table 2.** Factors considered in the factorial design of experiments of the extraction and stripping stages and respective levels.

| Stage | Factors | Unit | Levels | | |
|---|---|---|---|---|---|
| | | | Low (−1) | Standard (0) | High (+1) |
| Extraction | Equilibrium pH ($x_1$) * | dimensionless | 2.5 | 3.25 | 4.0 |
| | Organic to aqueous phase, O:A ($x_2$) | dimensionless | 0.5 | 1.25 | 2 |
| | Concentration of D2EHPA ($x_3$) | M | 0.4 | 0.5 | 0.6 |
| Stripping | Concentration of $H_2SO_4$ ($x_1$) | M | 0.05 | 1.025 | 2 |
| | Organic to aqueous phase, O:A ($x_2$) | dimensionless | 1 | 4.5 | 8 |
| | Stripping time ($x_3$) | min | 2 | 13.5 | 25 |

* Equilibrium pH after a contact time of 10 min, with a maximum variation of ±0.05 from the value defined in the design.

The process response, *y*, was used to fit the coefficients of a linear second-order regression model, using the linear least squares method. Only statistically significant variables were considered in the models (*p*-value smaller than the significance level of 0.05). Analysis of variance (ANOVA) was used to assess the significance of the fitted model. The variance of the response accounted for the models was evaluated by the coefficient of determination ($R^2$), and the existence of pure quadratic curvature was determined by hypothesis testing. Response surfaces and contour plots were used to assist the optimization of the processes.

## 3. Results and Discussion

### 3.1. Preliminary Tests of Extraction

Preliminary tests were performed to determine the best conditions to be further investigated in the factorial design of experiments. The extraction of Mn, Ni, Co, and Li at different contact times can be observed in Figure 1. The mechanism of extraction of manganese using D2HEPA is very fast. The extraction of Mn was about 60% after only 5 min of contact time, and after 10 min, the extraction achieves the maximum values (approximately 70%). The coextraction of Co, Ni, and Li is slightly higher after 5 min of contact time, but it is still lower than 20%. At 10 min of contact time, the increase in the extraction of Mn resulted in a decrease of the coextraction of the other metals. The coextraction of Co, Ni, and Li after 10 min of contact time was around 11, 5, and 3%, respectively. This is in accordance with the results reported in the literature. Chen et al. [24] studied the extraction of manganese from the leaching liquor of spent LIBs using cobalt-loaded D2EHPA, and they reported that the equilibrium was achieved after only 3 min. Hossain et al. [28] also observed that the kinetics of the manganese extraction using Co-D2EHPA was fast, and the equilibrium was achieved in 5 min. Thus, low contact times are required for the extraction of manganese.

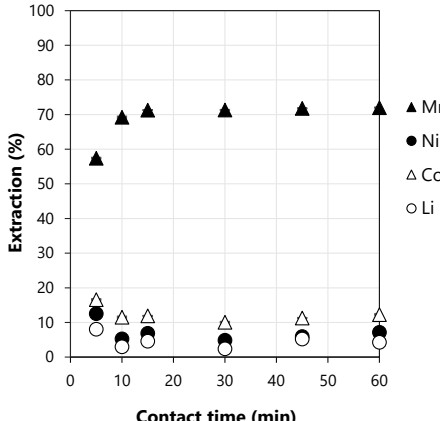

**Figure 1.** Extraction of metals at different leaching times. Conditions: O:A of 1:1; 0.5 M bis(2-ethylhexyl) phosphoric acid (D2EHPA) and pH of 3.5.

### 3.2. Effect of the Concentration of Modifier (% Volume of TBP)

Preliminary tests using TBP (tributyl phosphate, Sigma Aldrich, Germany) as a modifier were performed to evaluate its potential to increase the extraction of manganese as well as its separation from the other metals. The extraction of Mn, Ni, Co, and Li without using TBP and when volumetric concentrations of 2.5%, 5%, and 10% TBP were used can be seen in Figure 2, where the error bars represent the standard deviation of triplicates. The extraction of Mn had a slight increase when the concentration of TBP was increased until 5%. However, the coextraction of all other metals also increased when TBP was used as a modifier. For all evaluated metals, the extraction decreased when 10% of TBP was used. Considering that no formation of a third phase was observed, it was decided not to use TBP in the next tests.

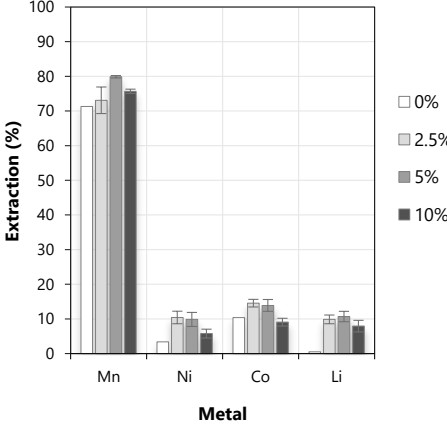

**Figure 2.** Extraction of metals using different volumetric concentrations of TBP as a phase modifier. Conditions: contact time of 10 min, 0.5 M D2EHPA, equilibrium pH of 3.5, organic to aqueous ratio (O:A) of 1:1. Error bars represent the standard deviation of triplicates.

### 3.3. Effect of the pH on the Extraction of Metals

The extraction of Mn, Co, Li, and Ni for three different molar concentrations of D2HEPA (0.4, 0.5, and 0.6 M) at different pH values can be seen in Figure 3. Some tests were performed using 0.2 M D2EHPA, but in this case, the extraction of manganese never exceeded 30%, and since this concentration is lower than the ones usually reported in the literature, further tests using 0.2 M D2EHPA were not performed. The initial pH of the synthetic solution based on the composition of the LIBs leach liquor was 3.8. After contacting the synthetic solution with the extractant, the pH of the aqueous phase decreased to about 2. This behavior was expected, considering the mechanism of extraction of metals

using D2EHPA (Equation (4)) described by Zhang and Cheng [14], which results in a decrease in the pH.

$$M^{2+} + 2\overline{(HA)_2} \leftrightarrows \overline{MA_4H_2} + 2H^+ \tag{4}$$

where $M$ represents the metal, $\overline{(HA)_2}$ represents D2EHPA in the organic phase, and $\overline{MA_4H_2}$ represents the metal–organic complex [14].

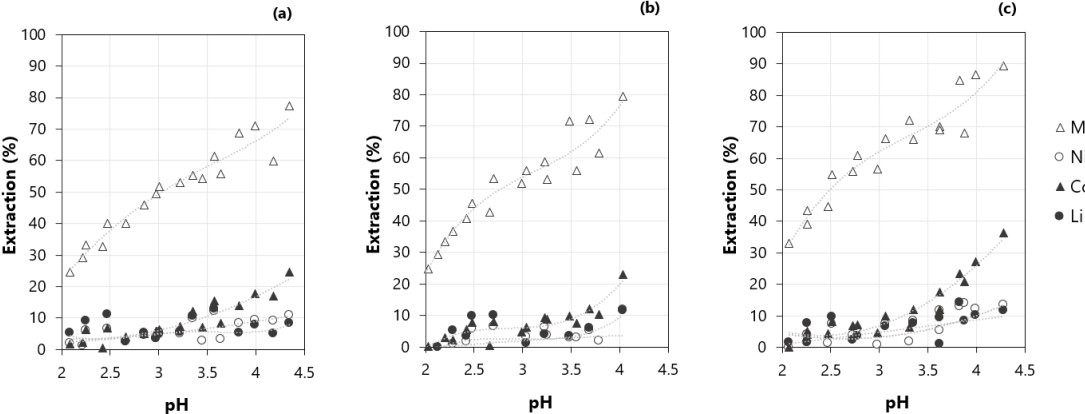

**Figure 3.** Extraction of metals using different molar concentrations of D2EHPA: (**a**) 0.4 M D2EHPA, (**b**) 0.5 M D2EHPA, (**c**) 0.6 M D2EHPA. Conditions: O:A of 1:1, contact time of 10 min.

The extraction of manganese increased with the pH for the three different concentrations of D2EHPA, but when the pH was increased to about 4, the coextraction of other metals was also more pronounced, mainly of cobalt. The increase in the molar concentration of D2HEPA also promoted an increase in the extraction of manganese, which was more pronounced when 0.6 M D2EHPA was used.

### 3.4. Effect of the Organic to Aqueous Ratio (O:A)

Preliminary tests were performed to evaluate the effect of the O:A ratio on the extraction of metals (Figure 4). The extraction of manganese increased with the O:A ratio (Figure 4a); however, the coextraction of cobalt also increased with the O:A ratio. For this reason, O:A ratios from 0.5 to 2 were further investigated in the factorial design of experiments. The isotherm representing the distribution of manganese in the aqueous and organic phase can be seen in Figure 4b. The extraction of manganese can be theoretically achieved after two extraction stages using an O:A ratio of 1.25.

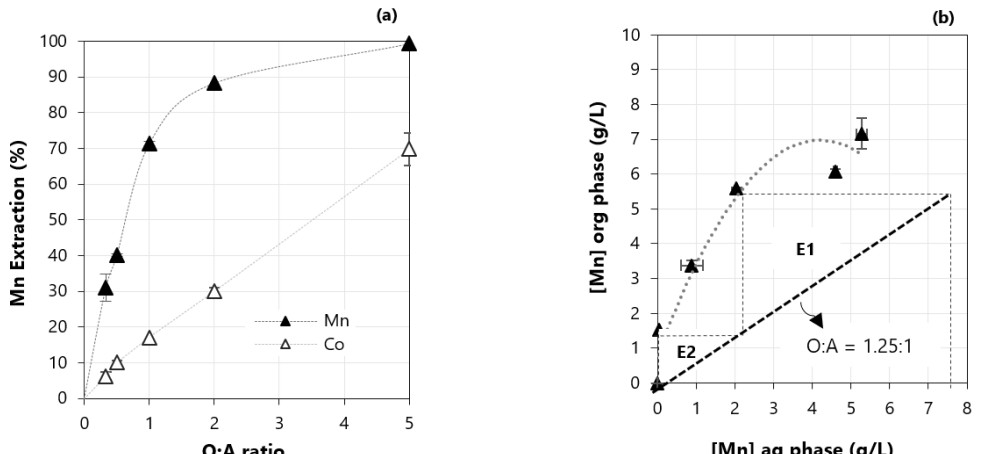

**Figure 4.** (**a**) Extraction of manganese and cobalt using different O:A ratios and (**b**) McCabe–Thiele diagram of the Mn extraction. Conditions: equilibrium pH of 3.5, 0.5 M D2EHPA, contact time of 10 min. Error bars represent the standard deviation of triplicates.

### 3.5. Extraction Stage: Factorial Design of Experiments and Regression Model

The conditions of the factorial design of experiments and respective responses (manganese and cobalt extraction) for each experiment are presented in Table 3. Tests from 1 to 8 correspond to the base $2^3$ design. Tests from 9 to 12 are the replicates in the central point of the design and were used to determine the experimental error. Tests from 13 to 18 are the axial points added to the design. All the tests were performed at room temperature using a contact time of 10 min. The concentrations of metals in the raffinate and in the organic phase are reported in the Supplementary Materials (Table S1), as well as the extraction of Ni and Li, which in general remain at low values. The Supplementary Material (Table S2) also reports the distribution ratios (*D*) and separation factors (β).

**Table 3.** Conditions of the experimental design and results for the extraction of manganese and cobalt.

| Run Order | Std Order | Coded Variables | | | Real Variables | | | Response (Extraction) | |
|---|---|---|---|---|---|---|---|---|---|
| | | $x_1$ | $x_2$ | $x_3$ | pH | O:A | D2EHPA | Mn (%) | Co (%) |
| 6 | 1 | −1 | −1 | −1 | 2.5 | 0.5 | 0.4 | 20 | 2 |
| 11 | 2 | 1 | −1 | −1 | 4 | 0.5 | 0.4 | 51 | 12 |
| 10 | 3 | −1 | 1 | −1 | 2.5 | 2 | 0.4 | 61 | 4 |
| 4 | 4 | 1 | 1 | −1 | 4 | 2 | 0.4 | 92 | 23 |
| 14 | 5 | −1 | −1 | 1 | 2.5 | 0.5 | 0.6 | 30 | 4 |
| 5 | 6 | 1 | −1 | 1 | 4 | 0.5 | 0.6 | 57 | 8 |
| 12 | 7 | −1 | 1 | 1 | 2.5 | 2 | 0.6 | 79 | 1 |
| 13 | 8 | 1 | 1 | 1 | 4 | 2 | 0.6 | 97 | 44 |
| 18 | 9 | 0 | 0 | 0 | 3.25 | 1.25 | 0.5 | 72 | 4 |
| 8 | 10 | 0 | 0 | 0 | 3.25 | 1.25 | 0.5 | 73 | 5 |
| 7 | 11 | 0 | 0 | 0 | 3.25 | 1.25 | 0.5 | 73 | 5 |
| 9 | 12 | 0 | 0 | 0 | 3.25 | 1.25 | 0.5 | 70 | 4 |
| 15 | 13 | −1 | 0 | 0 | 2.5 | 1.25 | 0.5 | 48 | 1 |
| 16 | 14 | 1 | 0 | 0 | 4 | 1.25 | 0.5 | 88 | 25 |
| 2 | 15 | 0 | −1 | 0 | 3.25 | 0.5 | 0.5 | 38 | 9 |
| 1 | 16 | 0 | 1 | 0 | 3.25 | 2 | 0.5 | 91 | 16 |
| 17 | 17 | 0 | 0 | −1 | 3.25 | 1.25 | 0.4 | 63 | 7 |
| 3 | 18 | 0 | 0 | 1 | 3.25 | 1.25 | 0.6 | 81 | 3 |

The adjusted regression model (*y*) for the extraction of manganese and the extraction of cobalt are represented by Equations (5) and (6), respectively. The models are only valid for the range of values tested in this study, and they only include factors with a statistically significant effect on the responses (α = 0.05).

$$Mn\ (\%) = 72.0 + 14.7\ x_1 + 22.3x_2 + 5.7x_3 - 7.4x_2^2 \tag{5}$$

$$Co\ (\%) = 6.6 + 10.0\ x_1 + 5.2x_2 + 6.0x_1x_2 + 3.7x_1x_2x_3 + 4.7x_1^2 \tag{6}$$

The results of the analysis of variance of the fitted models for the extraction of manganese and cobalt are presented in Table 4, which was adapted from the ANOVA table from the Regression Analysis tool of Excel (Analysis ToolPak add-in). The replicates in the central level of the design allow estimating the experimental pure error and decomposing the Residual Sum of Squares (RSS) into the Sum of Squares due to Pure Error (SSPE) and the Sum of Squares due to Lack of Fit (SSLOF). The presence of curvature was verified for both models using the pure curvature testing (*p*-value = 0.048 and 0.046 for manganese and cobalt, respectively). The significance of the fitted models is indicated by the results of the *F*-test. The model adequacy was assessed by the Lack of Fit (LOF) test, but the results were lower than the significance level (α = 0.05) for both models, given the low experimental error in the central point of the design and a small variance of the experimental error when compared to the residual error.

**Table 4.** Results of the analysis of variance of the fitted models for the extraction of manganese and cobalt.

| Response Source | | Degree of Freedom | Sum of Squares | Mean Square | F-Value | p-Value |
|---|---|---|---|---|---|---|
| Manganese extraction | Regression | 10 | 7964.8 | 796.5 | 43.6 | $2.4 \times 10^{-5}$ |
| | Residual | 7 | 127.9 | 18.3 | - | - |
| | Lack of fit | 4 | 120.4 | 30.1 | 12.2 | $3.4 \times 10^{-2}$ |
| | Pure error | 3 | 7.4 | 2.5 | - | - |
| | Totals | 17 | 8092.7 | - | - | - |
| Cobalt extraction | Regression | 10 | 1988.3 | 198.8 | 17.6 | $4.9 \times 10^{-4}$ |
| | Residual | 7 | 79.2 | 11.3 | - | - |
| | Lack of fit | 4 | 78.0 | 19.5 | 49.2 | $4.6 \times 10^{-3}$ |
| | Pure error | 3 | 1.2 | 0.4 | - | - |
| | Totals | 17 | 2067.5 | - | - | - |

Pareto charts of the standardized effects of the variables on the responses are presented in Figure 5a for the manganese extraction and in Figure 5b for the cobalt extraction. The standardized effects were calculated by dividing each coefficient by its standard error. The standardized effects correspond to the *t*-statistic values. A variable is considered statistically significant if its *p*-value is smaller than the defined significance level (0.05 for a confidence level of 95%). The significance level is identified in the graphs by dashed lines (2.36 at abscissa) and it corresponds to the 0.975 quartile in the Student´s distribution, with seven degrees of freedom (total number of estimated coefficients subtracted from the total number of experiments). Thus, the effect of variables and their interactions is more significant as they are to the right of the red dashed line.

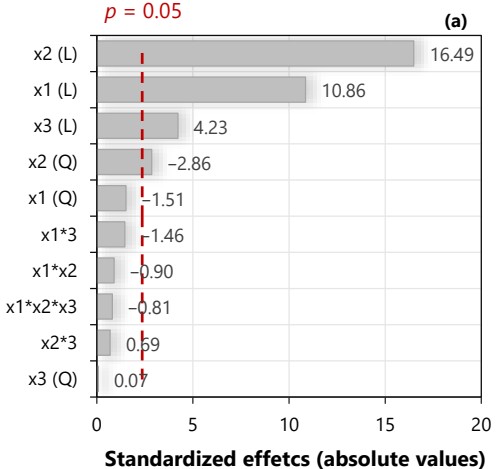
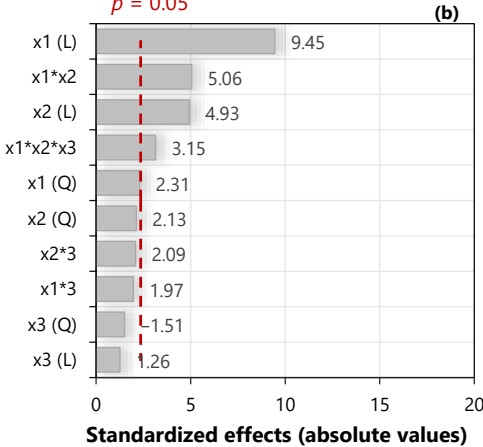

**Figure 5.** Pareto charts of the absolute values of the standardized effects of the factors for the regression model for (**a**) manganese extraction and (**b**) cobalt extraction with a significance level $\alpha = 0.05$. Legend: $x_1$: pH, $x_2$: O:A ratio, $x_3$: molar concentration of D2EHPA, (Q): quadratic terms, (L): linear terms.

The variables with higher effects on the manganese extraction were $x_2$ (O:A ratio), $x_1$ (pH), and $x_3$ (molar concentration of D2EHPA). The quadratic effect of the factor $x_2$ is also significant in the extraction of manganese. Then, it can be concluded that the extraction of manganese increases with the increase of the pH, extractant concentration, and the O:A ratio. The quadratic terms $x_1^2$ and $x_3^2$, as well as all the interactions, did not present a significant effect on the manganese extraction in the range of values tested in this work (at a confidence level of 95%).

Regarding the extraction of cobalt (Figure 5b), the main effects were accounted for the variables $x_1$ (pH), $x_2$ (O:A ratio) and the interactions of $x_1x_2$ and $x_1x_2x_3$, with a positive effect on the response with the increase of their levels. The quadratic terms $x_1^2$, $x_2^2$, and $x_3^2$, the factor $x_3$ (molar concentration of D2EHPA), as well as the interactions $x_1x_3$ and

$x_2x_3$ did not present a significant effect on the extraction of cobalt in the range of values considered for a confidence level of 95%.

The coefficient of determination ($R^2$) was used to assess the goodness of fit of the models. The model for the manganese extraction presented an $R^2 = 0.98$ and for the cobalt extraction an $R^2 = 0.96$. This coefficient indicates that 98% and 96% of the response variability is explained by the fitted models, respectively. The relation between the experimentally observed responses for the extraction of manganese (Figure 6a) and cobalt (Figure 6b) is represented in the scatter plots below. This relation demonstrates that the adjusted models can provide a good fit to the experimental results under the range of values considered in the study.

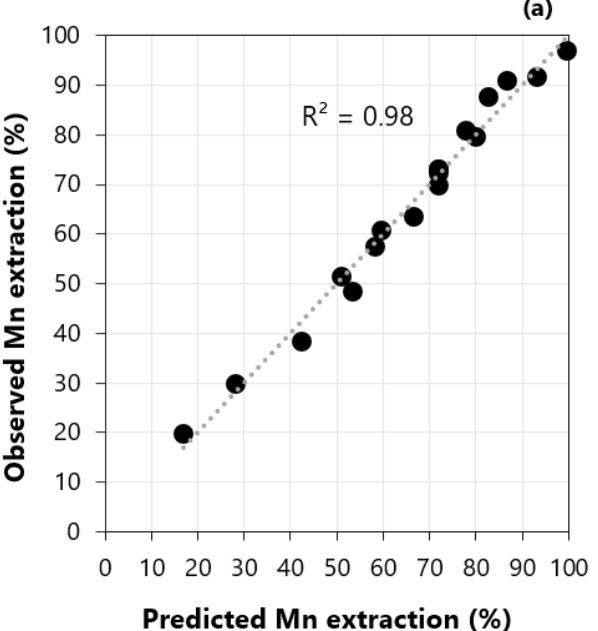 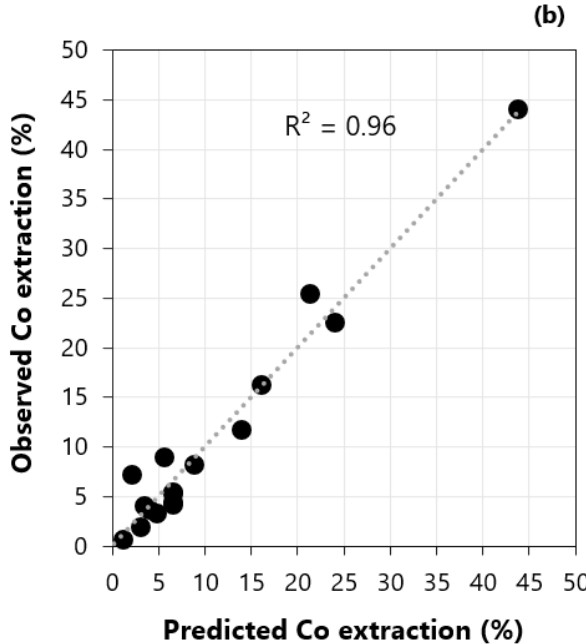

**Figure 6.** Responses predicted by the model versus experimentally observed: (**a**) manganese extraction and (**b**) cobalt extraction.

### 3.6. Response Surfaces: Extraction of Manganese and Cobalt

To help to understand the effect of the different factors on the extraction of manganese and cobalt, response surfaces were used. They were depicted using contour plots to show a clear representation of the surfaces. Contour plots are represented by a set of lines of constant response, being constructed in planes defined by pairs of variables. Therefore, each line represents a particular response of the fitted model.

The contour plots representing the manganese extraction when the factor $x_1$ (pH) was fixed at its low level ($-1$, pH = 2.5), standard level (0, pH = 3.2), and high level (+1, pH = 4) can be seen in Figure 7a–c, respectively. The responses for the extraction of cobalt under these same conditions are represented in Figure 7d–f. To construct the contour plots, the level of the factors $x_2$ (O:A ratio) and $x_3$ (molar concentration of D2EHPA) was changed from the low to the high level. The responses ($y$ = % extraction) are represented by legends on the left of each graph. Results are only valid in the range of values considered in this study.

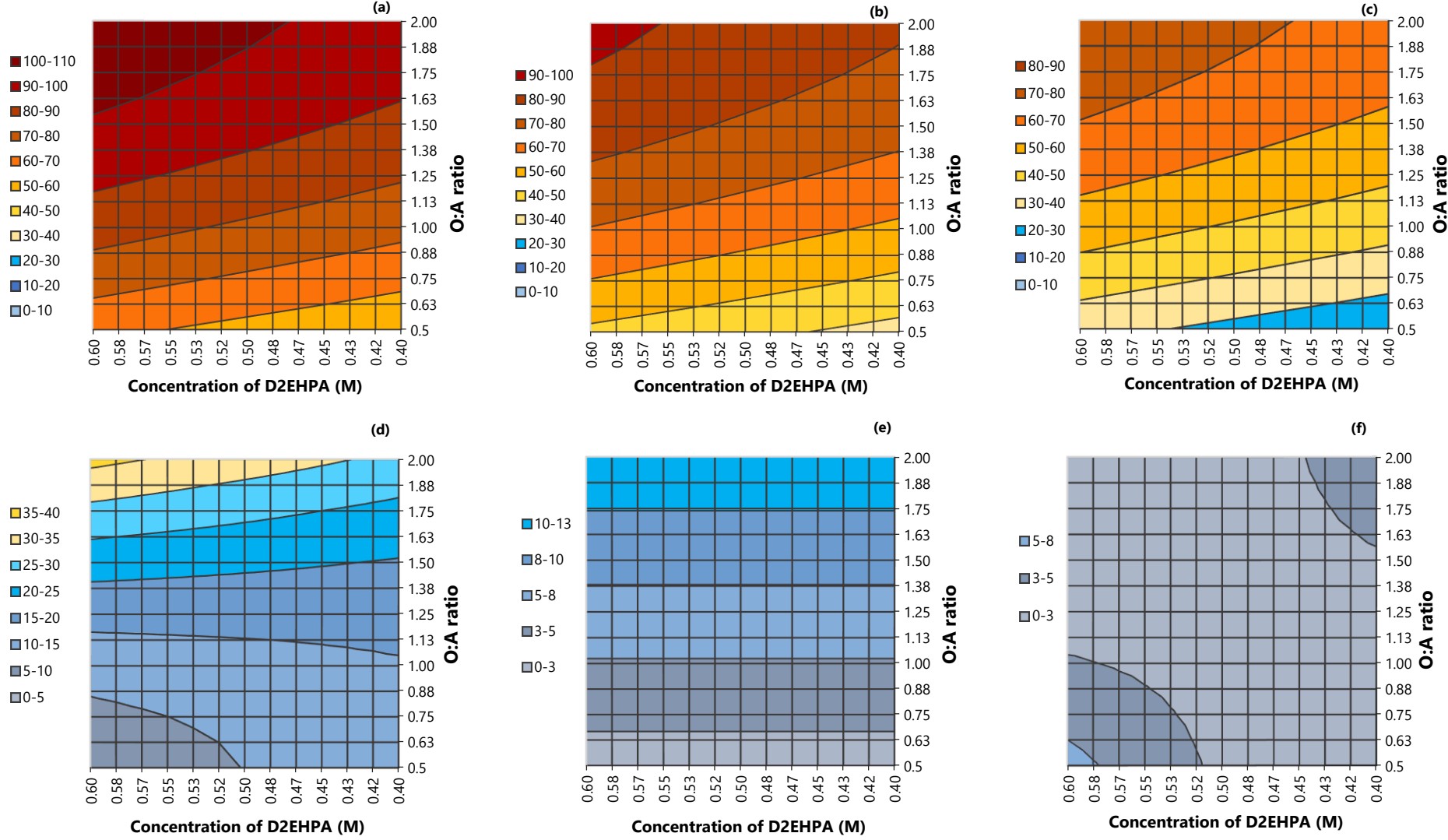

**Figure 7.** Contour plots representing the (**a–c**) extraction of manganese and the (**d–f**) coextraction of cobalt when the pH was set at 2.5 (**a,d**), at pH of 3.25 (**b,e**) and at pH 4 (**c,f**).

The extraction of manganese when the pH was set at 2.5 is represented in Figure 7a. High manganese extractions can be achieved for any level of concentration of D2EHPA provided that the O:A ratio is also at a high level, which is explained by the highest effect of the O:A ratio on the response. At the lowest pH, the lowest extraction of manganese was verified at the lowest level of the O:A ratio (0.5:1) and at the lowest concentration of extractant (0.4 M). On the other hand, when the pH was 2.5, the highest extraction of manganese was observed at the highest level of the O:A ratio (2:1) and at the highest level of concentration of D2EHPA (0.6 M). However, when the pH was 2.5, the extraction of manganese never exceeded 70–80%, which can be explained by the mechanism of the reaction of D2HPA, by which an increase in the concentration of $H^+$ ions will move the equilibrium to the left side, hiding the formation of products. When the pH was set at 2.5, it is possible to observe in Figure 7d that the extraction of cobalt was kept at a very low level and never exceeded 5%, which was reached only when high concentrations of D2EHPA or high O:A ratios were employed.

The behavior of the extraction of manganese when the pH was 2 was similar to the one when the pH was 3.25, as can be observed in Figure 7b. However, the increase in the pH resulted in an increase in the highest extraction of manganese, which was raised to 80–90%. The lowest extraction of manganese at pH 3.25 was also obtained when the concentration of D2EHPA and the O:A ratio were at their lowest levels (0.4 M and 0.5:1, respectively). The highest extraction of manganese at pH 3.25 was achieved when the other two factors were at the highest level (0.6 M and 2:1). Extractions of manganese above 70% can be obtained for the whole range of values tested for the concentration of D2EHPA, provided that the O:A ratio is at least 1.4:1. When the pH was set at the standard level (3.25), the extraction of cobalt is mainly dependent on the O:A ratio (Figure 7e). Thus, it is possible to keep the coextraction of cobalt below 8% provided that the O:A ratio does not exceed around 1.4:1.

Contour plots representing the extraction of manganese when the pH was set at 4 can be seen in Figure 7c. The extraction of manganese reached higher values when the other two factors were combined at a higher pH, which is explained by the significant effect of the pH on the response, as it was discussed in the regression analysis. At the highest pH, the extraction of manganese was always above 50%. The lowest extraction was obtained when the concentration of D2HEPA and the O:A ratio were at the lowest level (0.4 M and 0.5:1, respectively). When both factors were increased to the highest level, the extraction of manganese achieved the maximum results. It is important to highlight that for certain conditions, the fitted model slightly overestimated the responses (above 100%). The coextraction of cobalt also increased to higher values when the pH was set at the highest level (Figure 7f), which is also compatible with the significant effect of the pH on the cobalt response, which was observed in the regression analysis. The highest coextraction of cobalt was observed when the concentration of D2EHPA and the O:A ratio were at their highest levels (0.6 M and 2:1, respectively) and achieved around 35%. At pH 4, the coextraction of cobalt remained at lower levels when both the O:A ratio and concentration of D2EHPA were set at lower levels.

Considering the results using the fitted models, to keep the coextraction of cobalt low even though obtaining high extractions of manganese, the pH, O:A ratio, and concentration of D2EHPA should be kept at intermediate levels. For this reason, the next stages (scrubbing and stripping) were studied using a loaded organic obtained at the central level of the tested factors (pH of 3.25, O:A 1.25:1, and 0.5 M D2EHPA). The concentration of the loaded organic obtained at these conditions to be used in the next stages was compatible with the results of the factorial design of experiments.

### 3.7. Scrubbing of the Loaded Organic

According to Ritcey and Ashbrook [36], scrubbing usually refers to the removal of unwanted coextracted species in the loaded organic. The purpose of scrubbing the organic phase is to replace coextracted or mechanically entrained Co, Ni, or Li together with Mn [20]. Although it can be considered an important stage to purify the loaded organic

and selectively remove some undesired metals, the scrubbing stage was not studied in detail in this work, and the scrubbing conditions proposed by Peng et al. [20] were used. Thus, the loaded organic obtained using the standard conditions of the factorial design of experiments was scrubbed twice with a pure solution containing 4 g/L Mn prepared using $MnSO_4.H_2O$, without pH adjustment (pH: 4.4) for 10 min at an O:A ratio of 10:1. The final composition of the scrubbing solutions (1 and 2) after contact with the loaded organic and the resultant organic phase is presented in Table 5.

**Table 5.** Composition of the scrubbing solutions and the resultant organic phase after two scrubbing stages with 4 g/L Mn (O:A of 10:1, contact time of 10 min).

| Solution | Concentration (g/L) | | | |
|---|---|---|---|---|
| | **Mn** | **Co** | **Ni** | **Li** |
| Feed solution | 7.4 | 18.7 | 7.2 | 1.1 |
| Aqueous phase (after extraction) | 2.1 | 18.0 | 7.0 | 1.0 |
| Scrubbing solution 1 (aqueous phase) | 0.8 | 3.0 | 0.3 | 0.1 |
| Scrubbing solution 2 (aqueous phase) | 2.1 | 1.9 | <0.1 | <0.1 |
| Organic phase | 4.7 | 0.1 | 0.1 | <0.1 |

*3.8. Stripping Stage: Factorial Design of Experiments and Regression Model*

The experimental conditions of the factorial design for the stripping of the loaded organic and respective responses are presented in Table 6. The final concentrations of manganese and cobalt (g/L) in the stripping product were considered as the process responses. All experiments were performed at room temperature after two scrubbing stages (detailed in Section 3.7).

**Table 6.** Conditions of the experimental design and results for the stripping of cobalt and manganese.

| Random Order | Std Order | Coded Variables | | | Real Variables | | | Response | |
|---|---|---|---|---|---|---|---|---|---|
| | | $x_1$ | $x_2$ | $x_3$ | $[H_2SO_4]$ | O:A | Time | Mn (g/L) | Co (g/L) |
| 9 | 1 | −1 | −1 | −1 | 0.05 | 1 | 2 | 4 | 0.06 |
| 14 | 2 | 1 | −1 | −1 | 2 | 1 | 2 | 4 | 0.05 |
| 4 | 3 | −1 | 1 | −1 | 0.05 | 8 | 2 | 11 | 0.31 |
| 2 | 4 | 1 | 1 | −1 | 2 | 8 | 2 | 19 | 0.26 |
| 15 | 5 | −1 | −1 | 1 | 0.05 | 1 | 25 | 5 | 0.08 |
| 11 | 6 | 1 | −1 | 1 | 2 | 1 | 25 | 5 | 0.07 |
| 3 | 7 | −1 | 1 | 1 | 0.05 | 8 | 25 | 10 | 0.41 |
| 8 | 8 | 1 | 1 | 1 | 2 | 8 | 25 | 28 | 0.42 |
| 16 | 9 | 0 | 0 | 0 | 1.025 | 4.5 | 13.5 | 17 | 0.26 |
| 5 | 10 | 0 | 0 | 0 | 1.025 | 4.5 | 13.5 | 16 | 0.24 |
| 7 | 11 | 0 | 0 | 0 | 1.025 | 4.5 | 13.5 | 16 | 0.24 |
| 1 | 12 | 0 | 0 | 0 | 1.025 | 4.5 | 13.5 | 17 | 0.27 |
| 18 | 13 | −1 | 0 | 0 | 0.05 | 4.5 | 13.5 | 9 | 0.15 |
| 12 | 14 | 1 | 0 | 0 | 2 | 4.5 | 13.5 | 17 | 0.26 |
| 10 | 15 | 0 | −1 | 0 | 1.025 | 1 | 13.5 | 5 | 0.07 |
| 6 | 16 | 0 | 1 | 0 | 1.025 | 8 | 13.5 | 23 | 0.36 |
| 13 | 17 | 0 | 0 | −1 | 1.025 | 4.5 | 2 | 14 | 0.21 |
| 17 | 18 | 0 | 0 | 1 | 1.025 | 4.5 | 25 | 22 | 0.34 |

The regression models for the stripping of manganese and cobalt are represented by Equations (7) and (8), respectively, and only factors with a statistically significant effect on the responses were inserted in the models ($\alpha = 0.05$). The models are only valid for the range of values tested in this study.

$$Mn\ (g/L) = 16.9 + 3.4x_1 + 6.8x_2 + 2.0x_3 + 3.3x_1x_2 - 4.0x_1^2 - 3.1x_2^2 \qquad (7)$$

$$Co\ (g/L) = 0.25 + 0.14x_2 + 0.04x_3 + 0.03x_2x_3 \tag{8}$$

The results of the analysis of variance of the models are presented in Table 7. The presence of curvature was verified only for the model representing the manganese stripping with the pure curvature testing (*p*-value = 0.04). The results of the *F*-test can be related to the significance of the fitted models. The model adequacy was assessed by the LOF test, but the result for the manganese stripping was lower than the significance level ($\alpha = 0.05$), which can be related to the low experimental error in the central point of the design.

**Table 7.** Results of the analysis of variance of the fitted models for the stripping of manganese and cobalt.

| Response | Source | Degree of Freedom | Sum of Squares | Mean Square | *F*-Value | *p*-Value |
|---|---|---|---|---|---|---|
| Concentration of manganese | Regression | 10 | 880.2 | 88.0 | 20.4 | $3.0 \times 10^{-4}$ |
| | Residual | 7 | 30.2 | 4.3 | - | - |
| | Lack of fit | 4 | 28.7 | 7.2 | 13.9 | $2.8 \times 10^{-2}$ |
| | Pure error | 3 | 1.5 | 0.5 | - | - |
| | Totals | 17 | 910.4 | - | - | - |
| Concentration of cobalt | Regression | 10 | 0.2 | $2.4 \times 10^{-2}$ | 21.4 | $2.6 \times 10^{-4}$ |
| | Residual | 7 | $7.93 \times 10^{-3}$ | $1.1 \times 10^{-3}$ | - | - |
| | Lack of fit | 4 | $7.33 \times 10^{-3}$ | $1.8 \times 10^{-3}$ | 9.0 | $5.1 \times 10^{-2}$ |
| | Pure error | 3 | $6.08 \times 10^{-4}$ | $2.0 \times 10^{-4}$ | - | - |
| | Totals | 17 | 0.2 | - | - | - |

Pareto charts of the standardized effects of the variables on the responses are presented in Figure 8. A significant effect on the stripping of manganese (Figure 8b) was accounted for the three main variables: $x_1$ (concentration of H$_2$SO$_4$), $x_2$ (O:A ratio), and $x_3$ (stripping time). The interaction effect of $x_1$ and $x_2$ was also significant, as well as the effect of the quadratic term $x_1^2$. Thus, the stripping of manganese will increase with the increase of the levels of these three variables. The quadratic terms $x_2^2$ and $x_3^2$, as well as all the other interactions, did not have a significant effect on the manganese stripping, considering the range of values tested at a confidence level of 95%. Only the variables $x_2$ (O:A ratio) and $x_3$ (stripping time) had a positive and significant effect on the stripping of cobalt (Figure 8b). Thus, the concentration of acid did not show a significant effect on the stripping of cobalt in the tested range nor did it have all the interactions and quadratic terms (at a confidence level of 95%).

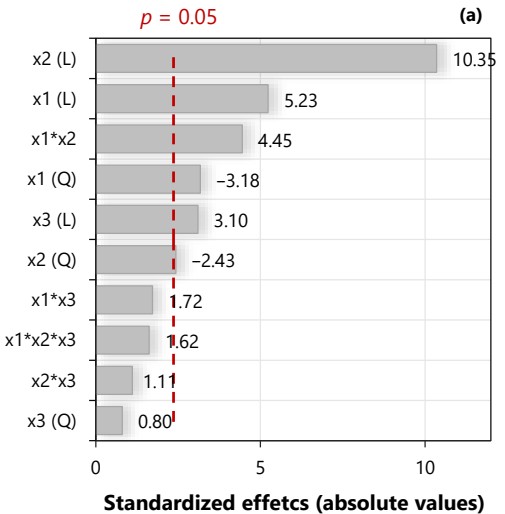
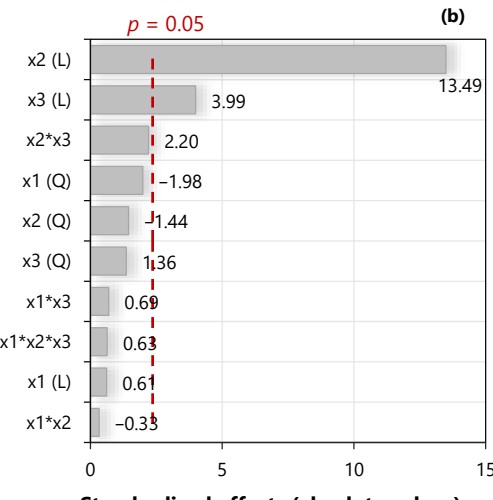

**Figure 8.** Pareto charts of the absolute values of the standardized effects of the factors for the regression model for the (**a**) manganese stripping and (**b**) for the cobalt stripping. Significance level $\alpha = 0.05$. Legend: $x_1$: molar concentration of H$_2$SO$_4$, $x_2$: O:A ratio, $x_3$: stripping time, (Q): quadratic terms, (L): linear terms.

Both models presented an $R^2 = 0.97$, which is indicative that a large proportion of the variance of the response can be explained by the independent variables, considering the range of values tested in the experiments. The relation between the experimentally observed responses and those obtained using the fitted model for the stripping of manganese and cobalt are represented in Figure 9a,b, respectively, which illustrates how the models provide a good fit to the experimental results.

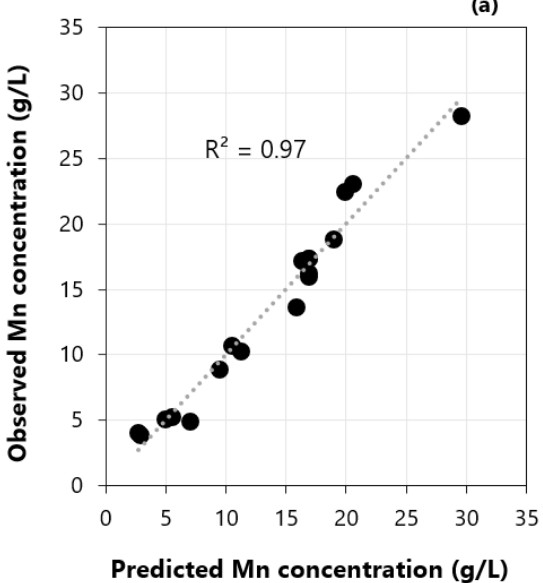
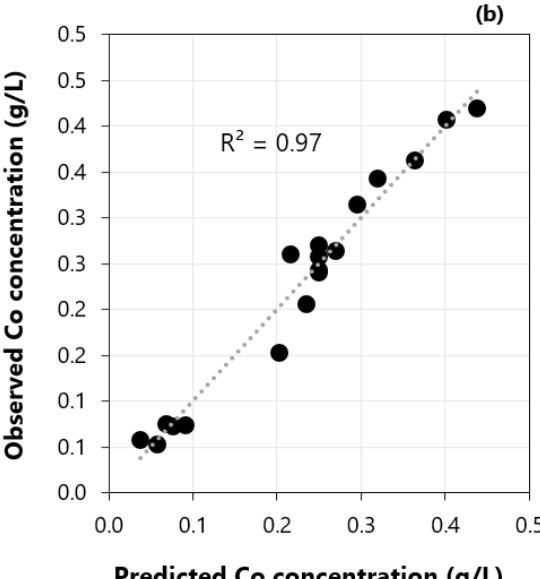

**Figure 9.** Responses predicted by the model versus experimentally observed: (**a**) manganese stripping and (**b**) cobalt stripping.

### 3.9. Response Surfaces: Stripping of Manganese and Cobalt

The contour plots in Figure 10a–c represent the response surfaces of the manganese stripping when the factor $x_2$ (O:A ratio) was set at its low level (−1, O:A = 1:1), standard level (0, O:A = 4.5:1), and high level (+1, O:A = 8:1), respectively. The stripping of cobalt for different combinations of O:A ratio and time is represented by the contour plots in Figure 10d, given that the concentration of sulfuric acid did not have a significant effect on it. The values of the response ($y$) are represented by legends on the left side of each graph. Results are only valid in the range of values considered in this study. The concentrations of metals remaining in the organic phase and in the stripping product for each test are reported in the Supplementary Materials (Table S3).

When the O:A used in the stripping was 1:1 (Figure 10a), a low concentration of manganese was obtained and never exceeded 10 g/L, which was expected given the larger volume of aqueous phase. At the lowest concentration of $H_2SO_4$ (0.05 M), the lowest concentration of manganese in the stripping product was verified at the lowest stripping time (2 min), being lower than 3 g/L Mn. With the increase in the concentration of $H_2SO_4$ and in the leaching time, a slight increase in the concentration of manganese was observed (maximum of 10 g/L).

The stripping behavior of manganese when the O:A ratio was set at 4.5:1 can be observed in Figure 10b. At this O:A ratio, the lowest concentration of manganese was around 8–10 g/L, and it was reached when the concentration of $H_2SO_4$ was the lowest (0.05 M) at the shortest stripping time (2 min). Increasing the concentration of acid from 0.9 to 2 M and the stripping time from 15 to 25 min promoted an increase in the concentration of manganese, which reached around 20 g/L.

The concentration of manganese was the highest when the O:A ratio was set at 8:1 (Figure 10c) and it was higher than 10 g/L for all tested conditions. The concentration of

manganese reached higher values when the other two variables (time and concentration of acid) were combined at the highest O:A ratio, which is related to the highly significant effect of the O:A ratio on the response, as previously discussed in the regression analysis. When the concentration of acid was at the lowest level (0.05 M) and the stripping time was also at the lowest level (2 min), the concentration of manganese was around 10 g/L. When both factors were increased to their highest levels, the concentration of manganese achieved the maximum results (23–25 g/L). In Figure 10a–c, it is also possible to observe how the concentration of acid ($x_1$) has a more pronounced effect on the concentration of manganese in the stripped product, which was also represented by a quadratic term in the model, causing a curvature in the response surface. Thus, a slight increase in the concentration of acid can cause a higher effect on the concentration of manganese.

The stripping of cobalt (Figure 10d) was mainly affected by the O:A ratio and by the leaching time, while the concentrations of $H_2SO_4$ tested in this study did not have a significant effect on the concentration of cobalt in the stripped liquor. The concentration of cobalt increased along with the O:A ratio and the stripping time, but it never exceeded 0.5 g/L. Thus, it can be concluded that very high concentrations of manganese in the stripping product (>23 g/L) can be obtained using high O:A ratios and concentrations of sulfuric acid of around 1 M. However, the stripping time should not exceed around 13 min, in order to keep the concentration of cobalt at a low level (<0.3 g/L). Additionally, the fitted models can support the optimization of the stripping process.

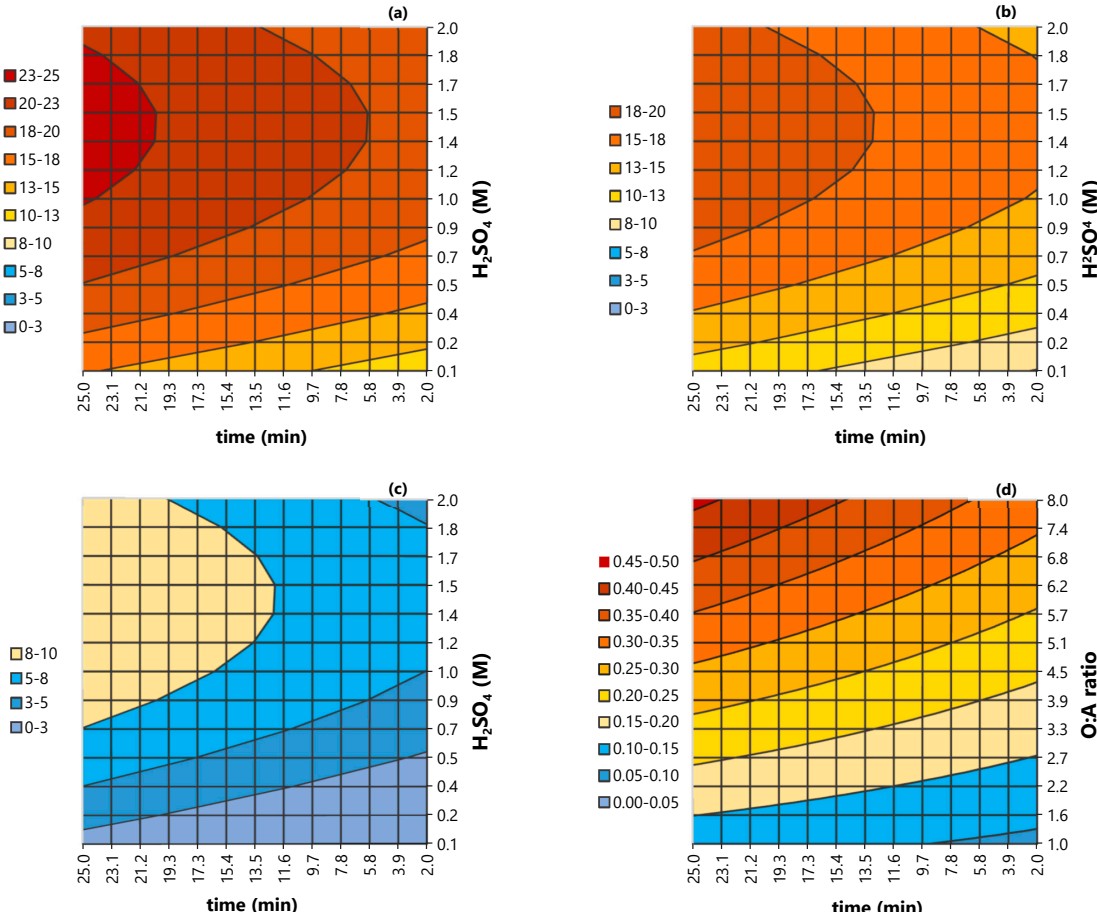

**Figure 10.** Contour plots representing the (**a**–**c**) stripping of manganese (**a**) when the O:A was set at 8:1, (**b**) when the O:A ratio was set at 4.5:1, and (**c**) when the O:A ratio was 1:1. (**d**) represents the stripping of cobalt at different combinations of stripping time and O:A ratios.

The fitted models can help to optimize the solvent extraction of manganese and can also assist with the construction of distribution isotherms and McCabe–Thiele diagrams, which are very helpful to predict the distribution of metals in both phases of the system (aqueous and organic) and to theoretically determine the number of required stages. The distribution isotherms for the stripping of manganese and cobalt, whose results were determined using the fitted models, are presented in the Supplementary Materials (Figure S1).

## 4. Conclusions

The recovery of manganese from a solution based on lithium-ion batteries was investigated using the factorial design of experiments and the response surface methodologies in order to assess the effect of different factors on the solvent extraction of manganese. These methodologies were also used to optimize the extraction and stripping stages, aiming to minimize the coextraction of cobalt. Preliminary tests were performed to determine the experimental conditions to be further investigated in the factorial design of experiments. The use of a modifier (TBP) was tested, but the formation of a third phase was not observed, and for this reason, additional tests with a modifier were not performed. The extraction of manganese using D2EHPA was fast, and maximum results were achieved after 10 min of contact time.

The factors evaluated in the extraction stage were the equilibrium pH, the molar concentration of D2EHPA, and the organic to aqueous ratio. Under optimized conditions (O:A of 1.25:1, pH 3.25, and 0.5 M D2EHPA), extractions above 70% Mn were reached in a single extraction stage with a coextraction of around only 5% Co, which was mostly removed in two scrubbing stages. Other combinations of factors can also result in high extractions of manganese and low coextractions of cobalt. In general, the coextraction of lithium and nickel remained low. The variables considered for the optimization of the stripping stage were the concentration of sulfuric acid, the organic to aqueous ratio, and the stripping time. A stripping product containing around 23 g/L Mn and around 0.3 g/L Co can be obtained under optimized conditions (O:A of 8:1, 1 M $H_2SO_4$, and around 13 min of contact time) in a single stripping stage. Increasing the number of extraction stages can promote an increase in the concentration of manganese loaded in the organic phase and should be further investigated in up-scale tests using mixer-settlers. Moreover, the fitted models for the extraction and stripping stages can help optimize these processes and can also assist with the construction of McCabe–Thiele diagrams to predict the number of stages required to maximize the recovery of manganese.

The results obtained can support further investigations on the recovery of manganese from spent lithium-ion battery solutions, which are an important secondary resource of manganese, using solvent extraction with D2EHPA. Moreover, the use of methodologies to model and optimize the process can assist the process management, considering that multiple combinations of factors can result in high extractions of manganese and low coextractions of other metals. Knowing these alternatives can help to better design the process to reduce the consumption of energy and reagents, minimizing costs and environmental impacts.

**Supplementary Materials:** The following are available online at https://www.mdpi.com/2075-4701/11/1/54/s1,Table S1. Conditions of the experimental design and concentrations of metals in the raffinate and in the organic phase after one extraction stage. Contact time of 10 min. Legend: [aq]: concentration of metal in aqueous phase, [org] concentration of meta in organic phase, Table S2. Conditions of the experimental design, distribution ratios (*D*) and separation factors (β) after one extraction stage. Contact time of 10 min, Table S3. Conditions of the experimental design and concentrations of metals remaining in the organic phase and in the stripping product. Legend: [aq]: concentration of metal in aqueous; phase, [org] concentration of metal in organic phase, Figure S1. Distribution isotherms of (**a**) manganese stripping and (**b**) cobalt stripping obtained using the fitted models. Conditions used as input in the fitted models: stripping time: 13.5 min (coded variable: 0), O:A ratio: 8:1 (coded variable: +1), concentration of $H_2SO_4$: 1 M (coded variable: 0).

**Author Contributions:** Data curation, formal analysis, investigation, visualization, writing—original draft preparation, N.V.; Conceptualization, methodology, validation, writing—review and editing, N.V., N.R., C.E., M.P.; resources, N.R., C.E., M.P.; project administration and supervision, M.P.; funding acquisition, C.E., M.P.; All authors have read and agreed to the published version of the manuscript.

**Funding:** This research was supported by VINNOVA (reference number of project: 2019-02069).

**Institutional Review Board Statement:** Not applicable.

**Informed Consent Statement:** Not applicable.

**Data Availability Statement:** The data presented in this study are available in supplementary material here.

**Conflicts of Interest:** The authors declare no conflict of interest.

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
