# Peer review of "Optimization of Manganese Recovery from a Solution Based on Lithium-Ion Batteries by Solvent Extraction with D2EHPA"

_metals, doi:10.3390/met11010054_

Round 1

Reviewer 1 Report

The manuscript is a study of manganese extraction with the comparison of cobalt extraction from used lithium-ion batteries. It was performed mainly by factorial design of experiments and response surface methodologies in order to assess the effects of various process parameters.

A few points need to be changed.  

  • The first two paragraphs at the beginning of introduction do not seem to be appropriate.
  • Line 89, ‘ investigated in detail in previous work’ requires literatures.
  • The value of pH for (a) and (d) in the caption of Fig. 7 is inconsistent with the text.
  • Line 164 needs to be corrected.
  • The conclusion is weak.

Reviewer 2 Report

This manuscript is easy to read, with a clear introduction that explains the objectives of the work and presents interesting results with a good mathematical interpretation. Therefore I will only suggest minor changes before publication. These are the following:

  • Line 50 I wasn’t able to reach reference 10 with the information provided.
  • Lines 87-89: A previous work is mentioned but not referenced, where probably the use of the synthetic solution is more clear. If the results are unpublished then say it clearly.
  • Line 164 There is an error (is it a reference to figure 1?)
  • Line 183. Please, give the menaning of the error bars presented on Figre 2… i.e. error bars are a standard deviation of triplicates… or whatever.
  • Line 200. This should probably be eq. 4. All the following equations should also increase their numbers.
  • Lines 215-227: The legend of figure 4 and the text of these lines require some revision. Also, on Figure 4 I would also suggest not to use shades behind the dark symbols of the figure. Also the meaning of the error bars is missing.
  • Line 235. The link given at the end of the manuscript (line 479) is not working (on my browser) so I couldn't see these Supplementary Materials nor those mentioned on line 420.
  • Line 247: … Residual Some of Squares ?
  • Line 299: Regarding Figs. 7 (and 10), they are easy to read and helpful for understanding the results. Nevertheless, if possible, it would be better that the same values present the same colour for all the figures 7a to 7f. I understand that the responses are quite different and this may be a difficulty to do what I suggest

Reviewer 3 Report

The reviewed manuscript describes modeling and optimizing the solvent extraction of manganese from a solution based on lithium-ion batteries with the use of factorial designs of experiments and the response surface methodology. Authors applied the specific approach here, for the first time, and reached extractions above 70% Mn in a single extraction stage with a coextraction of less than 5% Co, which was mostly removed in two scrubbing stages. Finally, obtained results can support further investigations focused on the recovery of manganese from spent LIBs.

I can confirm that the paper presents an useful research contribution to modern extraction processes. I found the presentation clear and correct, and the results should be of interest for a wide audience. The results can also help to get an important secondary resource of a critical material for many important industrial sectors. I also believe that the approach developed in this paper would be useful for the understanding of some other similar processes.

This paper is quite well written and also timely important. Results would be useful for many readers. The manuscript would then be suitable for publication. I am therefore inclined to recommend for publication of the paper in “Metals”.

Author Response

We thank you very much for reading the manuscript and for your positive feedback.